# Developing Conversion Factors of LCIA Methods for Comparison of LCA Results in the Construction Sector

**Yahong Dong** [1], **Md. Uzzal Hossain** [2], **Hongyang Li** [3,4,5] **and Peng Liu** [1,*]

[1] Qingdao Research Center for Green Development and Ecological Environment, Qingdao University of Science and Technology, No. 99 Songling Road, Qingdao 266061, China; yhdong@qust.edu.cn
[2] Department of Civil Engineering, The University of Hong Kong, Pokfulam Road, Hong Kong, China; uzzal@hku.hk
[3] Business School, Hohai University, Nanjing 211100, China; lihy@hhu.edu.cn
[4] School of Civil Engineering and Transportation, South China University of Technology, Guangzhou 510641, China
[5] State Key Laboratory of Subtropical Building Science, South China University of Technology, Guangzhou 510641, China
* Correspondence: pliu@qust.edu.cn

**Abstract:** The inconsistency caused by different life cycle impact assessment (LCIA) methods is a long-term challenge for the life cycle assessment (LCA) community. It is necessary to systematically analyze the differences caused by LCIA methods and facilitate the fair comparison of LCA results. This study proposes an effective method of conversion factors (CFs) for converting the results of 8 LCIA methods for 14 impact categories and then demonstrates its application in the construction sector. Correlation analyses of the datasets of construction materials are conducted to develop CFs for the impact categories. A set of conversion cards are devised to present the CFs and the associated correlation information for the LCIA methods. It is revealed that the differences between LCIA methods are largely caused by the characterization methods, rather than due to the metrics. A comparison based only on the same metrics but ignoring the underlying LCIA mechanisms is misleading. High correlations are observed for the impact categories of climate change, acidification, eutrophication, and resource depletion. The developed CFs and conversion cards can greatly help LCA practitioners in the fair comparison of LCA results from different LCIA methods. Case studies are conducted, and verify that by applying the CFs the seemingly incomparable results from different LCIA methods become comparable. The CF method addresses the inconsistency problem of LCIA methods in a practical manner and helps improve the comparability and reliability of LCA studies in the construction sector. Suggestions are provided for the further development of LCIA conversion factors.

**Keywords:** life cycle assessment; life cycle impact assessment; harmonization; comparison; construction

## 1. Introduction

Life cycle assessment (LCA) has been increasingly adopted to evaluate the environmental impacts of industrial products and complex systems. In accordance with ISO 14,040 [1], an LCA study should include four interdependent phases: (i) goal and scope definition, (ii) inventory analysis, (iii) impact assessment, and (iv) interpretation. The third phase, which is commonly known as life cycle impact assessment (LCIA), transforms the life cycle inventory (LCI) results into understandable indicators for the environmental impact categories [2], such as climate change, eutrophication and human toxicity. There are a variety of LCIA methods that have been developed over the past 30 years. Notable ones are CML [3], EDIP [4], ILCD [5], ReCiPe [6], and TRACI [7]. These LCIA methods often have varying impact categories, inventory classification, indicators, characterization models, temporal and spatial horizons, and normalization and weighting methods, which may lead to different LCA results [8,9].

The notable obstacle that the practitioners often encounter in comparing LCA results comes from the LCIA methods. For an impact category, the LCIA methods that have been applied in the case studies may provide different indicators, which seemingly make the comparison impossible. For instance, eutrophication can be indicated either by nitrogen or by phosphorus. Santo et al. [10] studied an office building in Netherlands and found that the eutrophication impact of the studied building is $2.27 \times 10^3$ kg N eq (1.19 kg N eq/m$^2$), whereas Soust-Verdaguer et al. [11] analyzed two single-family houses in Uruguay and reported that the eutrophication impact for the masonry house is 0.0039 kg PO$_4$ eq/m$^2$, and for the timber house is 0.0045 kg PO$_4$ eq/m$^2$. Similar problems also exist for other impact categories. Notably, acidification is estimated by sulfur dioxide or hydrogen ion; toxicity is estimated by 1,4 dichlorobenzene (1,4-DB), vinyl chloride (C$_2$H$_3$Cl), or comparative toxic units (CTUh), etc.

The inconsistencies caused by different LCIA methods are a longtime problem for the LCA community and was reported in almost every field of LCA applications [12–23]. In the construction sector, as an example, Monterio and Freire [24] compared the results of cumulative energy demand (CED), CML and Eco-indicator 99 for a single-family house, and found that the three methods are consistent in climate change, acidification and eutrophication, but inconsistent in photochemical oxidation, ecotoxicity, human toxicity, etc. Without a scientific method to convert different LCIA methods, the inconsistencies can lead to unfair comparisons and ultimately may decrease the reliability of LCA [25]. A company may pick an LCIA method that favors their own product for "green washing" purposes [26].

Great efforts were made to deal with the LCIA inconsistency problem. The international UNEP/SETAC Life Cycle Initiative launched a flagship project to provide a guidance and build consensus on LCIA indicators, trying to ensure consistency of indicator selection and assessments across impact categories [27,28]. Under the umbrella of Life Cycle Initiative, research on the harmonization of LCIA methods was carried out for several impact categories, including climate change [29], ecotoxicity [30], particulate matter [29], ecosystem [31], water consumption [32], and natural resources [20,29]. With so many aspects, such as impact categories, indicators, characterization modeling, and regional effects [33], to consider, the harmonization of LCIA methods is however a long-term task. On the other hand, the pursuing for scientific consensus on LCIA methods might hold back new method developments, and we may have to "agree that we disagree" [26].

The comparison of LCIA methods and the reporting of the problems remain important to the LCA community. The common practice is to convert the metrics of the characterization factors to be same and then compare the LCIA results based on the converted metrics. This is not fully justified because the contradictions in the underlying LCIA characterization models have not been uncovered. Sometimes, the metrics of the same indicator cannot be converted at all due to the differences in physical dimensions. However, there were progresses that might give hints toward a solution. It is noted that Lasvaux et al. [34] identified a simplified set of environmental indicators and concluded that only 4–6 dimensions are sufficient to explain at least 90–95% of the variance for each set of indicators. Steinmann et al. [35] conducted Principal Component Analysis for the indicators of a variety of LCIA methods and found that six indicators can cover the 92% variance. Esnouf et al. [36] developed an index to evaluate the accordance between LCI and LCIA, which indicates that the inventory of materials may affect the appropriateness of the characterization mechanisms. Thus, the spectra of materials are better to be considered together with the LCIA characterization models, and this should be valid for different industrial sectors.

The comparison of LCIA methods should be in depth, pointing to the underlying characterization mechanisms. Furthermore, the analysis should be systematic and have a sound mathematical basis. This study aims at developing a conversion method to convert the results from different LCIA methods to be comparable. We build a mathematical toolbox for converting results between LCIA methods, intended to facilitate fair comparisons,

as well as help identify existing problems in the underlying mechanisms. The statistical regression method is employed to analyze the LCIA outcomes and develop the conversion factors (CFs). A series of conversion cards are devised to present the CFs and associated correlation information for the LCIA methods. In this study, the CFs are employed for the construction sector, both for the urgent needs [37,38] and for a demonstration purpose.

## 2. Methodology

### 2.1. Research Design

As shown in Figure 1, to develop the conversion factors, we use linear regression to correlate the important construction materials and derive the conversion factors. The conversion factors between two LCIA methods are calculated for each impact category. Then, the conversion cards are developed for each impact category. Using the conversion cards, the results from different LCIA methods can be converted to the same indicator, so that the results be fairly compared.

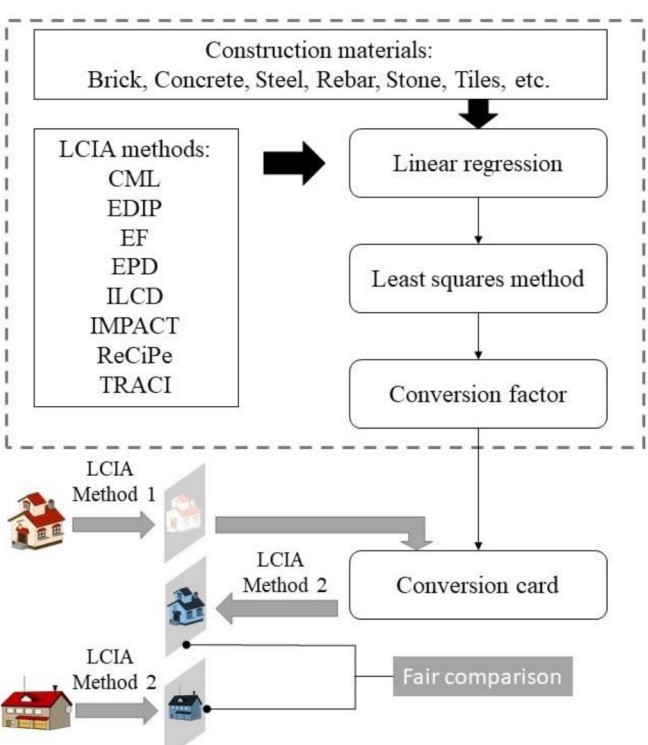

**Figure 1.** Illustration of the development of conversion factors and the application of conversion cards.

### 2.2. LCIA Methods and Impact Categories

According to the LCA standard ISO 14,044 [2], the LCIA phase consists of mandatory and optional elements. The mandatory elements include selection of impact categories, classification of LCI results, and characterization. The optional elements are normalization, grouping, weighting, etc. An LCIA method should at least include the mandatory elements, while it may provide optional elements. Since the optional elements such as normalization and weighting are processed on the characterization results, the different results of characterization models may be further distorted in these optional operations, making the LCIA method comparison less meaningful. In this study, we focus on the mandatory elements of LCIA to compare the characterization results from different LCIA methods, while leaving aside the vague analyses of the optional elements.

There are two approaches of characterization modeling—midpoint and endpoint. The midpoint approach (problem-oriented) evaluates the environmental impacts at the intermediate level along the cause-effect chain. The endpoint approach (damage-oriented) focuses on the final impact of pollutions on the Areas of Protection (AoPs), i.e., human

health, ecosystem, and resources. For example, the midpoint indicator of global warming is the emissions of greenhouse gases, and its endpoint indicator is Disability of Adjusted Life Years (DALY) for human health and Potentially Disappeared Fraction (PDF) for ecosystem [39]. Midpoint approach is more comprehensive, while endpoint approach is more concise [40]. Classical LCIA methods either use a midpoint approach or an endpoint approach for the characterization modeling; a few recent LCIA packages may provide both. This study mainly focuses on midpoint LCIA methods, since the endpoint approach is less transparent as stated above and accordingly has lower reliability [37].

The selection of LCIA methods is based on the following criteria: (i) the method is available in commercial software, such as SimaPro [41] or GaBi [42], as this facilitates a wide range of applications; (ii) the method involves multiple impact categories; and (iii) the latest version is studied, not the superseded version. According to these criteria, eight LCIA methods are analyzed, including CML, EPD, EDIP, EF, ILCD, IMPACT2002+, ReCiPe, and TRACI (see Table 1 for details). The impact categories are selected mainly depending on their availability in the selected LCIA methods. Most of the impact categories available in the LCIA methods are studied, except that an impact category is provided only by one LCIA method and its comparison cannot be carried out. Fourteen impact categories are analyzed in this study and shown in Table 1.

**Table 1.** Selected LCIA methods and impact categories. Metrics of impact categories are shown for each LCIA method.

| LCIA Methods | CML | EDIP | EF | EPD | ILCD | IMPACT | ReCiPe | TRACI |
|---|---|---|---|---|---|---|---|---|
| **References** | [3] | [4] | [43] | environdec.com (accessed on 2 April 2021) | [5] | [44] | [6] | [7] |
| **Region** | Europe | Europe | Europe | Global | Europe | Europe | Global | North America |
| **Version** | IA-baseline | 2003 | 2.0 | 2018 | 2001 Midpoint+ | 2002+ | 2016 Midpoint(H) | 2.1 |
| **Approach** | Mid | Mid | Mid/End | Mid | Mid | Mid/End | Mid | Mid |
| **Global warming** | kg $CO_2$ eq | kg $CO_2$ eq | kg $CO_2$ eq | kg $CO_2$ eq | kg $CO_2$ eq | kg $CO_2$ eq | kg $CO_2$ eq | kg $CO_2$ eq |
| **Acidification** | kg $SO_2$ eq | $m^2$ | mol H+ eq | kg $SO_2$ eq | mol H$^+$ eq | kg $SO_2$ eq | kg $SO_2$ eq | kg $SO_2$ eq |
| **Ozone depletion** | kg CFC-11 eq | kg CFC-11 eq | kg CFC-11 eq | kg CFC-11 eq | kg CFC-11 eq | kg CFC-11 eq | kg CFC-11 eq | kg CFC-11 eq |
| **Eutrophication** | kg $PO_4$ eq | kg P | kg P eq | kg $PO_4$ eq | kg P eq | kg $PO_4$ P-lim | kg P eq | kg N eq |
| **Energy consumption** | MJ | | MJ | MJ | | MJ primary | kg oil eq | MJ surplus |
| **Resource** | kg Sb eq | PR2004 | kg Sb eq | kg Sb eq | kg Sb eq | | kg Cu eq | |
| **Smog** | kg $C_2H_4$ eq | per.ppm.h | kg NMVOC eq | kg NMVOC | kg NMVOC eq | kg $C_2H_4$ eq | kg $NO_x$ eq | kg $O_3$ eq |
| **Water depletion** | | | $m^3$ depriv. | $m^3$ eq | $m^3$ water eq | | $m^3$ | |
| **Human toxicity (Cancer)** | kg 1,4-DB eq | person | CTUh | | CTUh | kg $C_2H_3Cl$ eq | kg 1,4-DCB | CTUh |
| **Human toxicity (Non-Cancer)** | kg 1,4-DB eq | person | CTUh | | CTUh | kg $C_2H_3Cl$ eq | kg 1,4-DCB | CTUh |
| **Particulate matter** | | | disease inc. | | kg PM2.5 eq | kg PM2.5 eq | kg PM2.5 eq | kg PM2.5 eq |
| **Ecotoxicity (Freshwater)** | kg 1,4-DB eq | $m^3$ | CTUe | | CTUe | kg TEG water | kg 1,4-DCB | CTUe |
| **Land use** | | | Pt | | kg C deficit | $m^2$org.arable | $m^2$a crop eq | |
| **Ionizing radiation** | | | kBq U-235 eq | | k Bq U235 eq | Bq C-14 eq | kBq Co-60 eq | |

Note: Mid: midpoint approach; End: endpoint approach. EF and IMPACT2002+ include midpoint and endpoint indicators for different impact categories. "eq" refers to equivalent.

### 2.3. Datasets for Comparison

To calculate the conversion factors of the aforementioned LCIA methods, datasets of materials were needed as model inputs. In this study, 14 construction materials were selected and the datasets were retrieved from the Ecoinvent database [45] (Table 2). The selected construction materials should cover a large span of the material spectrum so that they appropriately represent the construction sector as an important field of LCA application. According to [46], the selected construction materials can represent 98.5% of the total environmental impacts of all building materials.

**Table 2.** Selected datasets of construction materials in Ecoinvent database.

| Material | Dataset in Ecoinvent | FU |
|---|---|---|
| **Asphalt** | Mastic asphalt GLO丨 market for 丨 Conseq, S | 1 kg |
| **Brick** | Clay brick GLO丨 market for 丨 Conseq, S | 1 kg |
| **Cement** | Cement, blast furnace slag 5–25%, US only RoW丨 market for 丨 Conseq, S | 1 kg |
| **Concrete** | Concrete, 35 MPa GLO丨 market for 丨 Conseq, S | 1 kg * |
| **Door** | Door, inner, wood GLO丨 market for 丨 Conseq, S | 1 kg * |
| **Fiber** | Cellulose fibre, inclusive blowing in GLO丨 market for 丨 Conseq, S | 1 kg |
| **Glass** | Flat glass, uncoated GLO丨 market for 丨 Conseq, S | 1 kg |
| **Mortar** | Lime mortar GLO丨 market for 丨 Conseq, S | 1 kg |
| **Plaster** | Cover plaster, mineral GLO丨 market for 丨 Conseq, S | 1 kg |
| **Rebar** | Reinforcing steel GLO丨 market for 丨 Conseq, S | 1 kg |
| **Steel** | Steel, low-alloyed GLO丨 market for 丨 Conseq, S | 1 kg |
| **Stone** | Natural stone plate, cut GLO丨 market for 丨 Conseq, S | 1 kg |
| **Tiles** | Ceramic tile CH丨 production 丨 Conseq, S | 1 kg |
| **Window frame** | Window frame, aluminium, U = 1.6 W*m$^{-2}$ K GLO丨 market for 丨 Conseq, S | 0.1 kg * |

* Functional units (FU) are changed for these datasets, to guarantee LCIA results comparable. According to the documentation in the Ecoinvent database, density of concrete is 2315 kg/m$^3$; density of door is 27.6 kg/m$^2$; density of window frame is 50.7 kg/m$^2$.

### 2.4. Development of Conversion Factors

The least squares method is used in regression analysis to minimize the sum of squared residuals between an observed value and a fitted value. The conversion factors (CFs) of an impact category between two LCIA methods are developed based on regression analysis using the least squares method. The conversion of the impact assessment result from the ith to the jth LCIA method is simply given by a linear function:

$$F_l(i, j; k) = CF_l(i, j) \, IAR_l(i; k). \tag{1}$$

In Equation (1), $IAR_l(i; k)$ is the impact assessment result of the lth impact category from the ith LCIA method that is contributed by the kth material. $CF_l(i, j)$ is the conversion factor between the ith LCIA method and the jth LCIA method for the lth impact category. Here, the converted quantity $F_l(i, j; k)$ corresponds to $IAR_l(j; k)$, the impact assessment result of the lth impact category from the jth LCIA method that is contributed by the kth material. In this study, there are 14 impact categories within 8 LCIA methods (referring to Table 1), so $l \in (1,14)$ and $i,j \in (1,8)$. We have 14 materials studied (referring to Table 2), thus $k \in (1,14)$. The sum of the squared residuals is

$$S_l(i, j) = \sum_{k=1}^{14} [IAR_l(j; k) - F_l(i, j; k)]^2. \tag{2}$$

To minimize the squared residuals, the gradient of $S_l(i, j)$ with respect to $CF_l(i, j)$ is set to be zero, i.e.,

$$\frac{dS_l(i, j)}{dCF_l(i, j)} = 0. \tag{3}$$

This solves $CF_l(i, j)$ and we get Equation (4):

$$CF_l(i, j) = \frac{\sum_{k=1}^{14}[IAR_l(i; \ k)IRA_l(j; \ k)]}{\sum_{k=1}^{14} IAR_l(i; \ k)^2} \tag{4}$$

This is the explicit form of the conversion factor from the ith to the jth LCIA method for the lth impact category. In order to understand the quality of the conversion of results between two LCIA methods, the associated correlation coefficients or R-squared of the lth impact category are also evaluated by:

$$R_l^2 = \frac{\sum_{k=1}^{14}[IAR_l(j; \ k) - F_l(i, j; \ k)]^2}{\sum_{k=1}^{14}\left[IAR_l(j; k) - \frac{1}{14}\sum_{k=1}^{14} IAR_l(j; k)\right]^2} \tag{5}$$

## 3. Results

### 3.1. Conversion Cards

For each impact category, a conversion card is devised to show the conversion factors between individual LCIA methods. There are totally 14 conversion cards, as shown in Figure 2. The vertical axis of a conversion card represents the ith LCIA methods, and the horizontal axis represents the jth LCIA methods. The conversion factors (CFs) are given as the cell data in conversion cards. For example, the environmental impact of 1 kg cement to acidification is evaluated to be 0.0485 kg $SO_2$ eq in CML. By substituting the conversion factor $F_2(1, 2) = 15.67$ into Equation (1), we get the estimated result for the EDIP method,

$$F_2(1, 2; \ 3) = CF_2(1, 2) \ IAR_2(1; \ 3) = 15.67 \times 0.0485 = 0.75 \ m^2 \tag{6}$$

which is very close to the result directly computed from EDIP (0.75 $m^2$).

The $R^2$ of LCIA method conversions are coded as cell colors. Referring to the color bar in Figure 2, yellow represents a greater $R^2$, meaning high correlation, while red represents a less $R^2$, meaning low correlation. For those cells colored in yellow, the CFs can be adopted when results from different LCIA methods are compared. On the contrary, for those cells colored in red, the CFs should be used with caution or not be used.

In the following sub-sections, the CFs in the conversion cards are further discussed, in particular, for those in red cells. It should be noted that some CFs in red cells can still be utilized if proper adjustments are made on the regression models. Details will be provided subsequently, and data are provided in Supplementary Materials.

### 3.2. Climate Change (Global Warming)

The selected LCIA methods use the same indicator (global warming potential, GWP) and metric (kg $CO_2$ eq) for climate change (or global warming). The CFs show that the LCIA methods are consistent for climate change, except ILCD. It is found that the $R^2$ is very low (0.068~0.072) for the regression models of ILCD. The inconsistency is caused by two materials, i.e., door and fiber, which have 'carbon dioxide, air (input from nature)' in the inventory. ILCD method provides a characterization factor with a negative value of '−1' for 'carbon dioxide, air (input from nature)', while this characterization factor is absent in other LCIA methods. As a result, for datasets containing 'carbon dioxide, air (input from nature)', the results of climate change in ILCA cannot be compared with other methods.

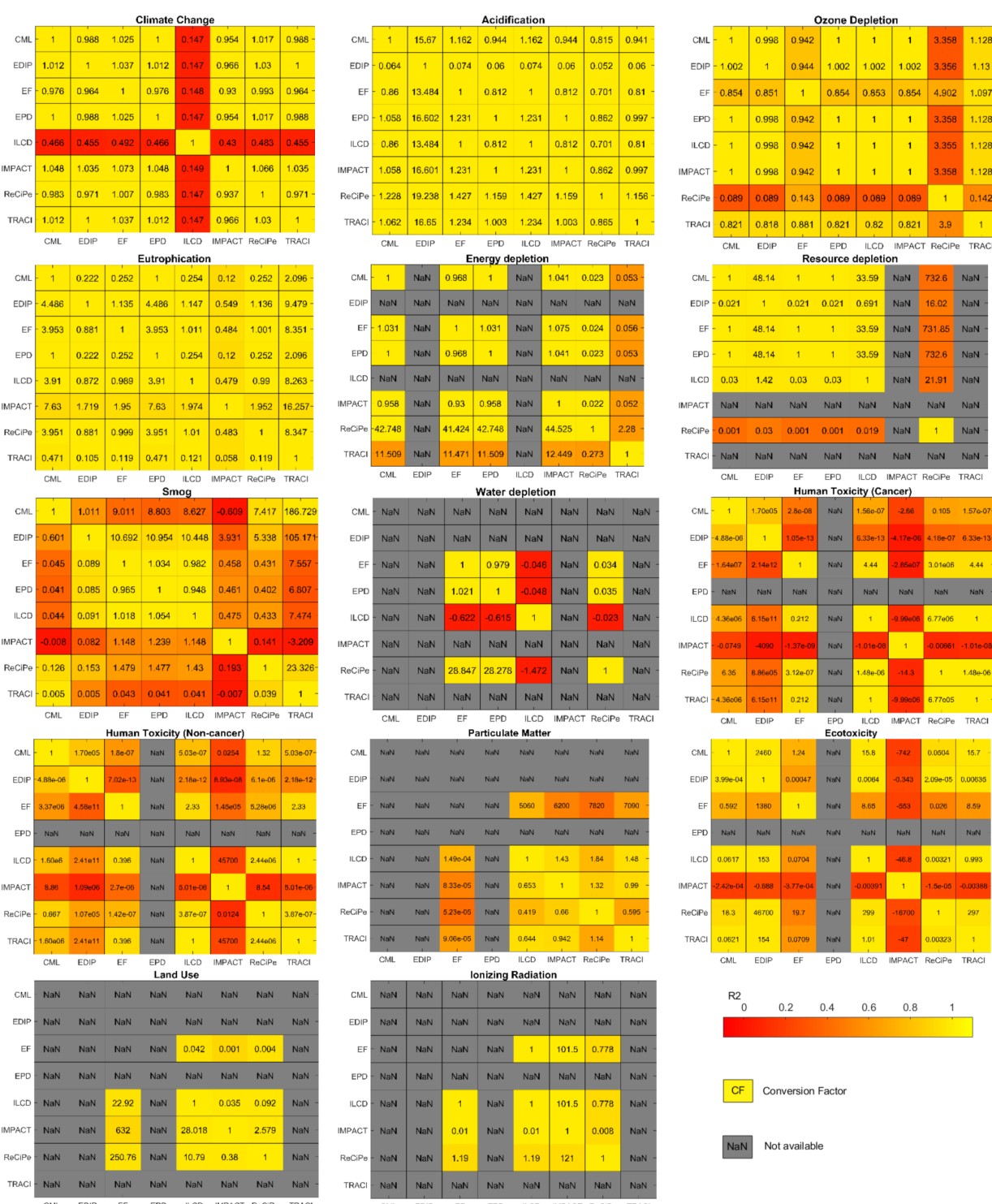

**Figure 2.** Conversion cards of impact categories. Yellow: high correlation; Orange: low correlation; Red: not correlated; Gray: CFs not available; NaN: not a number. (Refer to Table 1 for the metrics of LCIA methods.).

### 3.3. Acidification

Different indicators and metrics are used for acidification: CML, EPD, IMPACT2002+, ReCiPe and TRACI adopt acidification potential expressed in kg $SO_2$ eq; EF and ILCD adopt accumulated exceedance expressed in mol H+ eq; and EDIP uses the area of ecosystem that exceeds the critical load of acidification expressed in $m^2$. The selected LCIA methods are correlated and the $R^2$ are as high as 0.99. For the method IMPACT2002+,

there are two impact categories related to acidification, namely 'Aquatic acidification' and 'Terrestrial acidification/nutrient'. In the conversion card, 'Aquatic acidification' is used, in accordance with most LCIA methods. The results of 'Terrestrial acidification/nutrient' of IMPACT2002+ are further analyzed by conducting the regression model with results of 'Aquatic acidification'. The resulting $CF_2(i, j)$ is 2.81 (where $i$ is for 'Aquatic acidification' and $j$ is for 'Terrestrial acidification/nutrient'), and the $R^2$ is 0.99. For instance, 1 kg of brick releases 0.00103 kg $SO_2$ eq for 'Aquatic acidification' and 0.00452 kg $SO_2$ eq for 'Terrestrial acidification/nutrient'. It is noted some LCIA methods, e.g., ReCiPe, also provide 'Terrestrial acidification' expressed in kg $SO_2$ eq. However, the results of 'Terrestrial acidification' from ReCiPe vary from results of 'Terrestrial acidification/nutrient' from IMPACT2002+, instead are very similar to the results of 'Aquatic acidification' from IMPACT2002+. Consequently, inappropriate comparison can be made if the results of 'Terrestrial acidification' from ReCiPe are compared to 'Terrestrial acidification/nutrient' from IMPACT2002+ without applying the conversion factors.

### 3.4. Ozone Depletion

For ozone depletion, the indicators and metrics are the same all across LCIA methods (indicator: ozone depletion potential, metric: kg CFC-11 eq). High correlations ($R^2$~0.99) are observed in most of the LCIA methods, including CML, EDIP, EPD, ILCD, and IMPACT2002+. The $R^2$ values related to EF are about 0.80, and the $R^2$ values related to TRACI are around 0.92. ReCiPe has the lowest $R^2$ values, which are only 0.29 with most of the LCIA methods, 0.55 with TRACI, and 0.70 with EF. The low correlation of ReCiPe with other LCIA methods is attributed to the characterization factor of dinitrogen monoxide ($N_2O$). ReCiPe provides the characterization factor of dinitrogen monoxide for the ozone depletion category, of which the value is 0.0011 kg CFC-11/kg $N_2O$. By examining the datasets, it is found that cement is an outlier in the correlation analysis. This is caused by the use of alternative fuel/material in the clinker production, in particular the meat and bone which consume soybeans, which in turn absorbed nitrogen fertilizer in its upstream processes. As a result, the ozone depletion impact is estimated as a negative value ($-4.9 \times 10^{-7}$ kg CFC-11/kg cement). Another outlier in the regression models of ReCiPe is fiber, which also contributes to characterization factor of dinitrogen monoxide. For EF and TRACI, cement and fiber are also the outliers leading to inconsistency. The difference between EF/TRACI and other LCIA methods is mainly attributed to the characterization factors of Halon-1301 (EF: 15.2; TRACI: 16; others: 12 kg CFC-11/kg), CFC-12 (EF: 0.73; others: 1 kg CFC/kg) and HCFC-12 (EF: 0.034; others: 0.05 kg CFC-11/kg).

Therefore, it is not recommended to compare the results of ozone depletion from ReCiPe and EF with other LCIA methods when the inventory of the studied product contains dinitrogen monoxide, in particular for the datasets similar to fiber and cement (as selected in this study). Adjustments are made by removing the two outliers and giving new CFs of ReCiPe, EF and TRACI with other LCIA methods. For example, the adjusted $CF_3(i, j)$ of CML ($i$) and ReCiPe ($j$) is 5.987 with $R^2$ of 0.95. The adjusted $CF_3(i, j)$ of CML ($i$) and EF ($j$) is 1.193 with R2 of 0.99. The adjusted $CF_3(i, j)$ of CML ($i$) and TRACI ($j$) is 1.298 with $R^2$ of 0.99. It is worth to notice that although the indicators and metrics are same between ReCiPe and other methods, the results of ozone depletion vary significantly (e.g., $2.72 \times 10^{-8}$ kg CFC-11/kg brick by CML and $8.22 \times 10^{-8}$ kg CFC-11/kg brick by ReCiPe). It implies that the direct comparison between LCIA results without considering the underlying model difference is misleading.

### 3.5. Eutrophication

The indicators and metrics of eutrophication are in general consistent in the LCIA methods. The impacts of eutrophication are assessed by the release of nutrients to environment. Either phosphorus (P) or nitrogen (N) can be used as an indicator of eutrophication. As eutrophication is a complex phenomenon which can happen in freshwater, marine water and soil, there are four LCIA methods trying to consider the variants of impact category of

eutrophication. EDIP provides terrestrial eutrophication ($m^2$) and aquatic eutrophication (kg N eq, kg P eq). EF provides freshwater eutrophication (kg P eq), marine eutrophication (kg N eq), and terrestrial eutrophication (mol N eq). ILCD includes freshwater eutrophication (kg P eg), marine eutrophication (kg N eq), and terrestrial eutrophication (mol N eq). ReCiPe provides freshwater eutrophication (kg P eq) and marine eutrophication (kg N eq). In the conversion card, the freshwater eutrophication is selected for comparison.

According to the conversion card of eutrophication, the $R^2$ are as high as 0.99 among LCIA methods. One exception is IMPACT2002+, of which $R^2$ is around 0.91–0.95. This is caused by the characterization factors in eutrophication of IMPACT2002+ which only includes phosphorus-related emissions, such as phosphoric acid and phosphate, but excludes nitrogen-related emissions. Consequently, for the datasets (e.g., cellulous fiber) with processes related to nitrogen emissions or absorptions, the results of eutrophication by IMPACT2002+ are not consistent with other methods. The missing characterization factors of eutrophication in IMPACT2002+ are the reason why the results of eutrophication in IMPACT2002+ are much lower than other LCIA methods.

### 3.6. Energy Depletion

Six LCIA methods provide the assessment on energy depletion but adopt different impact category names, such as abiotic depletion (fossil fuel), resource use, and fossil resource scarcity. Most of the LCIA methods use MJ as the metrics, except ReCiPe, which adopts kg oil eq. According to the conversion card, most of the LCIA methods are consistent in energy depletion (with high $R^2 \sim 0.99$), except TRACI ($R^2 \sim 0.65$). It is found that the outlier of regression models of TRACI is cement. The dataset of cement contains cementitious material of ground granulated blast furnace slag (GGBFS), of which the upstream processes include crude oil and hard coal. The characterization factors of the two substances in TRACI have different ratios as compared to other LCIA methods. In CML, the ratio is 2.32 (43.2 MJ/kg oil/18.16 MJ/kg coal), but in TRACI the ratio is 40 (6.6 MJ/kg oil/0.165 MJ/kg coal). If cement is removed, the adjusted $CF_5(i,j)$ of CML (i) and TRACI (j) is 0.0782 with $R^2$ of 0.99.

### 3.7. Resource Depletion

Resource depletion is accounted for by the amount of metal depleted. CML, EF, EPD and ILCD adopt antimony (Sb) as the indicators, while EDIP uses pure resource (PR) and ReCiPe uses copper (Cu). CML, EF and EPD generate exact same results, which can be clearly seen from the conversion card (values of CFs equal to 1). It is noted that ILCD, despite using the same metric with other methods, generates totally different results. This is because the impact category in ILCD also accounts fossil and renewable resources, leading to much higher values of the LCIA results. Although the metric of EDIP is different from other methods, its $R^2$ are as high as 0.99. ReCiPe is an obvious exception in the conversion card, with the $R^2$ only about 0.4. The inconsistency in ReCiPe is unfortunately not caused by certain characterization factors. In fact, the entire list of characterization factors of this impact category in ReCiPe are not correlated with those of other LCIA methods. It can be seen by conducting a comparison between the characterization factors of CML and ReCiPe. As a result, no adjustment can be made on the regression models to rescue the situation.

### 3.8. Smog

LCIA methods evaluate the impact of smog (caused by photochemical oxidation) using different indicators. CML and IMPACT2002+ adopts ethlylene (kg $C_2H_4$ eq); EF, EPD and ILCD use non-methane volatile organic compounds (kg NMVOC eq); ReCiPe adopts nitrogen oxides (kg $NO_x$ eq); and TRACI uses ozone (kg $O_3$ eq). The conversion card shows that EF, EPD, and ILCD are correlated with each other with high $R^2$ of 0.99. CML, ReCiPe and TRACI are in general correlated with $R^2$ of 0.90–0.98. However, IMPACT2002+ is not correlated with other LCIA methods. By examining the regression models, it is found that fiber is the outlier causing the inconsistency. The waste paper processing in the

upstream generates NMVOCs. CML and TRACI do not provide characterization factor of NMVOCs, but other methods do. If this outlier is eliminated in the other LCIA methods, the correlations can be recovered (referring to Supplementary Material).

### 3.9. Water Depletion

Four LCIA methods provide assessment on water depletion, namely EF, EPD, ILCD and ReCiPe, with Cubic meter ($m^3$) of water as the metric. EF, EPD and ReCiPe are correlated with high $R^2$ of 0.99. Results from EF and EPD are almost the same. Although ReCiPe is correlated with EF and EPD, the results from ReCiPe are much lower than from EF and EPD. For example, the manufacturing of 1 kg of asphalt consumes 0.21 $m^3$ water as estimated by EF and EPD, while it is only 0.0095 by ReCiPe. This leads to the conversion factor of EF and ReCiPe to be 0.034. ILCD is not correlated with other methods. For example, the $R^2$ of the regression analysis of ILCD and EF is as low as 0.017.

### 3.10. Human Toxicity (Cancer)

Seven LCIA methods provide assessments on human toxicity. EF, ILCD, IMPACT2002+, ReCiPe and TRACI evaluate human toxicity in two impact categories, i.e., carcinogens and non-carcinogens, while CML and EDIP provide a single impact category of human toxicity. CML and ReCiPe share the same indicator 1,4 dichlorobenzene (kg 1,4-DB eq). EF, ILCD and TRACI use comparative toxicity unit for humans (CTUh). EDIP uses the number of persons exposed to the air-borne emissions (person). IMPACT2002+ uses emission of chloroethylene (kg $C_2H_3Cl$ eq). According to the conversion card of Human toxicity (Cancer), EF, ILCD, TRACI and ReCiPe are correlated with $R^2$ of 0.94–0.99. It is observed that although CML and ReCiPe share the same metric of the indicator, whereas the results from CML is much larger than ReCiPe. The similar problem is observed for EF and ILCD/TRACI. Although the three methods share the same metric, the results from EF are significantly larger. The conversion card shows that CML, EDIP and IMPACT2002+ are not correlated. The inconsistency existing in CML, EDIP and IMPACT2002+ cannot be ascribed to any outlier, so no adjustment can be made to regression models for a remedy. In other words, the results of human toxicity (cancer) by these three methods cannot be compared with other methods.

### 3.11. Human Toxicity (Non-Cancer)

The indicators of human toxicity (non-cancer) are the same as human toxicity (cancer) for the LCIA methods. In the conversion card, TRACI, ILCD and ReCiPe are correlated, with $R^2$ of 0.94–0.99. EF is correlated with TRACI and ILCD ($R^2 \sim$ 0.92). The $R^2$ of CML with EDIP, TRACI, ILCD and ReCiPe are 0.80–0.88. Low correlations with others are observed in IMPACT2002+, with $R^2$ of 0.1–0.4. According to the analysis, reasonable comparison of human toxicity (non-cancer) can only be conducted between TRACI, ReCiPe and ILCD.

### 3.12. Particulate Matter

There are five LCIA methods providing the assessment of particulate matter. IMPACT2002+, ILCD, ReCiPe and TRACI adopt emission of $PM_{2.5}$ as the indicator, while EF uses disease incidence due to kg PM2.5 as the indicator. IMPACT2002+, ILCD and TRACI are in general correlated, with $R^2$ of 0.93–0.95. The correlation between ReCiPe and IMPACT2002+ is relatively low with $R^2$ of 0.87, while ReCiPe has even lower correlations with other LCIA methods ($R^2 \sim$ 0.41–0.77). Low correlations are also found for EF. As a result, particulate matter results from IMPACT2002+, ILCD and TRACI can be compared directly. Despite the same metric of $PM_{2.5}$ adopted, results of particulate matter from ReCiPe cannot be compared with others. Moreover, adjustments cannot be made for this impact category, as the low correlation is caused by the inconsistent characterization factors rather than certain processes.

### 3.13. Ecotoxicity

Different indicators and metrics for the ecotoxicity impact category are employed in the LCIA methods. CML and ReCiPe use 1,4 dichlorobenzene (kg 1,4-DB eq). EF, ILCD and TRACI use CTUh. EDIP uses volume of exposed compartment ($m^3$). IMPACT2002+ adopts triethylene glycol equivalents into water (kg TEG eq). According to the conversion card, high $R^2$ is observed for most of the LCIA methods (0.96–0.99), except IMPACT2002+ and EF. By examining the regression models of IMPACT2002+, an outlier is identified, i.e., fiber, whose upstream processes include recycling sludge from pulp and paper. Waste of pulp and paper can release aluminum into soil and IMPACT2002+ provides a characterization factor of 'aluminum, soil'. However, other methods do not provide such a characterization factor. If the outlier is removed, the adjusted correlation between CML and IMPACT2002+ would have $R^2 \sim 0.99$. EF has low correlations with other LCIA methods. An outlier of cement is identified with EF with a negative value of ecotoxicity. This is due to the zinc emitted in spoil from hard coal mining and scrap copper. By examining the characterization factors between EF and ILCD, it is found that most of the characterization factors are the same in the two methods, whereas the characterization factors of 'zinc, groundwater' and 'copper, groundwater' in EF are 0. However, the characterization factors of the sub-compartment of groundwater are not provided in ILCD, which instead adopts characterization factors of $3.86 \times 10^4$ CTUe/kg 'zinc, water' for 'zinc, groundwater' and $5.52 \times 10^4$ CTUe/kg 'copper, water' for 'copper, groundwater'. This leads to high values of results of cement in ecotoxicity by ILCD as compared to EF. If cement is removed in the regression model of EF and ILCD, the $R^2$ should be 0.99.

### 3.14. Land Use

Four LCIA methods provide assessment on land use. EF adopts an endpoint approach and represents land use in pt (point) based on the soil quality index. ILCD measures changes of soil organic matter in kg $C/m^2/a$. IMPACT2002+ evaluates the area of organic arable land. ReCiPe evaluates area of crop land (equivalent). Despite different metrics are adopted, the results from the four LCIA methods are correlated with $R^2$ of 0.91–0.99. Attention should be paid to IMPACT2002+ and ReCiPe. Both have $m^2$ of land area as metric, while the results from ReCiPe are about 2.5-fold that of IMPACT2002+.

### 3.15. Ionizing Radiation

Different indicators and metrics are adopted for ionizing radiation. EF and ILCD quantify the impact of ionizing radiation on the population in comparison with Uranium 235 (kBq U235 eq). IMPACT2002+ evaluates Bq C-14 eq for ionizing radiation. ReCiPe measures in comparison with Cobalt-60 (kBq Co-60 eq). The conversion card shows that the four methods are correlated, with $R^2$ of 0.92–0.99.

## 4. Case Studies

### 4.1. Aircretes

In this section, we shall illustrate how to use the LCIA method conversion factors developed in this paper in LCA practice. The case study is mainly for the demonstration purpose, rather than for the specifications of manufacturing details of the products. The LCIA results of two types of autoclaved aerated concrete (aircrete) products are obtained from two LCIA methods, respectively. The LCA comparison of the two aircretes is shown in Table 3. Aircrete I is autoclaved aerated concrete retrieved from Environmental Product Declarations (EPD) of a Turkish organization, and CML was adopted to evaluate its environmental impacts. Aircrete II is autoclaved aerated concrete block given by a dataset in Ecoinvent, and the environmental impact results are subsequently generated by TRACI. As shown in Table 3, the impact indicators of eutrophication, smog and energy depletion are not comparable for the two products, if the results of the two LCIA methods are directly placed side by side. For example, Aircrete I has an energy consumption of 1298 MJ, whereas the energy depletion of Aircrete II is estimated to be 64.9 MJ surplus.

**Table 3.** Comparison of LCIA results for two types of aircrete products, based on the developed conversion factors.

| Item | Aircrete I | Aircrete II | |
|---|---|---|---|
| Source | https://epdturkey.org (accessed on 15 April 2021) | Ecoinvent | |
| Data description | Autoclaved Aerated Concrete | Autoclaved aerated concrete block CH\| production \| Cut-off, U | |
| FU | 1 m$^3$ | 1 m$^3$ | |
| Boundary | Cradle-to-gate | Cradle-to-gate | |
| LCIA method | CML | TRACI | Converted to CML * |
| Climate change | 196 kg $CO_2$ eq | 137 kg $CO_2$ eq | 139 kg $CO_2$ eq |
| Acidification | 0.441 kg $SO_2$ eq | 0.272 kg $SO_2$ eq | 0.289 kg $SO_2$ eq |
| Ozone depletion | $8.13 \times 10^{-6}$ kg CFC-11 eq | $7.82 \times 10^{-6}$ kg CFC-11 eq | $6.42 \times 10^{-6}$ kg CFC-11 eq |
| Eutrophication | 0.127 kg $PO_4$ eq | 0.153 kg N eq | 0.0721 kg $PO_4$ eq |
| Smog | 0.0269 kg $C_2H_4$ eq | 4.51 kg $O_3$ eq | 0.0226 kg $C_2H_4$ eq |
| Energy depletion | 1298 MJ | 64.9 MJ surplus | 823 MJ |

* LCIA results of Aircrete II are converted to CML results by using conversion factors.

### 4.2. Buildings

Two building cases are compared using the developed conversion cards. The first case is an office building located in the Netherlands and has an area of 1900 m$^2$ with 800 m$^2$ for the ground floor and 1100 m$^2$ for the first floor [10]. The foundations and stairs are made of reinforced concrete and the floors are made of prefabricated hollow core slabs. The service life of the office building is 50 years. The cradle-to-grave life cycle stages, including product, transportation, maintenance and replacement, operational energy, and end of life are considered.

The second case is a prefabricated and transportable housing unit, the so-called living laboratory, located in Shanghai, China [47]. The building structure is a new shipping container, which is made of steel. The living laboratory provides a bedroom, office space, and a bathroom. The life span is 25 years for this temporary house. The cradle-to-grave life cycle stages, including pre-use stage (product, transportation and construction), use stage (operational energy, water use, maintenance and replacement of building materials) and end of life stage (deconstruction, waste processing, recycling/reuse and disposal) are analyzed.

As shown in Table 4, the LCA results of the office building were calculated using TRACI in the reference. On the other hand, the living laboratory was analyzed using ReCiPe. The direct comparison between the two studies is not possible due to the inconsistent indicators. Using the developed conversion cards, the ReCiPe results of the living laboratory are converted to TRACI, so that the two studies can be compared. For example, for the impact category of Smog, the living laboratory emits 410 kg $O_3$ eq/m$^2$, which is slightly larger than 392 kg $O_3$ eq/m$^2$ of the office building. By applying the conversion factors developed in this paper, the LCIA results are reasonably converted into the same metrics, making the LCA comparison possible.

**Table 4.** Comparison of LCIA results of two buildings, based on the developed conversion factors.

| Item | Office Building | Living Laboratory (Baseline) | |
|---|---|---|---|
| Source | [10] | [47] | |
| Location | The Netherlands | China | |
| Area | 1900 m$^2$ | 27 m$^2$ | |
| Structure | Reinforced concrete | Shipping container | |
| FU | m$^2$ | m$^2$ | |
| Boundary | Cradle-to-grave | Cradle-to-grave | |
| LCIA method | TRACI | ReCiPe | Converted to TRACI * |
| Climate change | 4473 kg $CO_2$ eq | 7759 kg $CO_2$ eq | 7532 kg $CO_2$ eq |
| Acidification | 32.8 kg $SO_2$ eq | 30.5 kg $SO_2$ eq | 35.2 kg $SO_2$ eq |
| Ozone depletion | $8.6 \times 10^{-6}$ kg CFC-11 eq | 0.00189 kg CFC-11 eq | $3.9 \times 10^{-4}$ kg CFC-11 eq |
| Eutrophication | 1.19 kg N eq | 1.72 kg P eq | 14.4 kg N eq |
| Smog | 392 kg $O_3$ eq | 17.8 kg $NO_x$ eq | 410 kg $O_3$ eq |
| Energy depletion | 51,263 MJ surplus | 1389 kg oil eq | 4589 MJ surplus |

* LCIA results of the living laboratory are converted to TRACI results by using conversion factors.

## 5. Discussion

### 5.1. Applications

It is recommended to apply the conversion factors (CFs) together with the conversion cards when LCA results from different studies are compared. In the case that the $R^2$ is low (orange and red colors in the conversion card cell), the adjusted conversion factors can be used. LCA practitioners are suggested to examine the LCI results and the version of LCIA method before using the CFs. It is also necessary to validate the converted results with their original results before using the converted results for comparison. For example, the original results should be presented as shown in Table 3 in their comparison scenario. In the case that the impact category of an LCIA method is not included in this study, it is suggested to calculate the CFs using Equation (4).

### 5.2. Limitations

This study develops a series of conversion cards to convert the LCIA results from different LCIA methods. There are several limitations of this study. First, we developed the conversion factors using the least squares method based on linear regression. However, there are errors (or residuals) for the prediction. Therefore, referring to the $R^2$ before the conversion factors are used is strongly recommended. Second, we included eight LCIA methods, while there are more available LCIA methods. In addition, the superseded methods are not considered but some of these methods are still used in LCA studies. Moreover, the 14 impact categories cannot cover all the available impact categories. Future studies are necessary to include more LCIA methods, the superseded methods, and also to encompass more impact categories.

## 6. Concluding Remarks

Fair comparisons between LCIA methods are a challenge in LCA. This study provides a practical method to compare the results from different LCIA methods and demonstrates its application in the construction sector. The developed conversion factors (CFs), together with the conversion cards, can greatly help LCA practitioners to perform comparison between LCIA methods. The case studies verify that by applying the CFs, the seemingly incomparable results from different LCIA methods can be comparable. The analysis of conversion factors can contribute to ease the inconsistency problem and enhance the reliability of LCA. The method of developing the conversion factors also can help LCIA developers to identify where the inconsistency is.

Key findings are summarized as follows:

a. The differences in the results from LCIA methods are caused by the characterization factors, rather than the metrics.

b.    Although the same metrics are adopted for some LCIA methods, a fair comparison cannot be guaranteed. A comparison based only on the same metrics but ignoring the underlying mechanisms is misleading.

c.    A small difference in characterization factors of LCIA methods can generate entirely different results, consequently leading to the uncorrelation between LCIA methods.

d.    High correlations are observed for climate change, acidification, eutrophication, and resource depletion.

e.    For some impact categories, the inconsistency is caused by certain characterization factors, such as climate change of ILCD, smog of ReCiPe, etc. In such cases, adjustments of regression models can be made to facilitate the comparison.

f.    Despite different metrics are adopted in acidification, the LCIA methods are highly correlated.

g.    Some impact categories cannot be compared, since the entire list of characterization factors are not correlated, such as human toxicity of IMPACT2002+, CML and EDIP.

Future work can be conducted to develop new conversion factors for other LCIA methods in addition to the eight methods studied in this paper and to involve more impact categories. Conversion factors should be developed for other industrial sectors, such as electronics, energy, transport, food, etc.

**Supplementary Materials:** The following are available online at https://www.mdpi.com/article/10.3390/su13169016/s1. The conversion factors and $R^2$ of the 14 impact categories are provided in an Excel file. Readers are suggested to refer to the first worksheet "Readme" before using the conversion factors.

**Author Contributions:** Conceptualization, Y.D. and P.L.; methodology, Y.D.; software, Y.D.; validation, M.U.H. and P.L.; investigation, Y.D. and P.L.; resources, H.L.; data curation, Y.D.; writing—original draft preparation, Y.D.; writing—review and editing, M.U.H., H.L. and P.L.; supervision, P.L.; funding acquisition, Y.D. and H.L. All authors have read and agreed to the published version of the manuscript.

**Funding:** This study is supported by the Start-up Funding of Qingdao University of Science and Technology (Grant No. 010029060), the Fundamental Research Funds for the Central Universities (B210201014), the "13th Five-Year" Plan of Philosophy and Social Sciences of Guangdong Province (2019 General Project) (Project No. GD19CGL27), and the State Key Laboratory of Subtropical Building Science, South China University of Technology, China (2020ZB17).

**Institutional Review Board Statement:** Not applicable.

**Informed Consent Statement:** Not applicable.

**Data Availability Statement:** The data presented in this study are available in the supplementary material.

**Conflicts of Interest:** The authors declare no conflict of interest.

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
