# Peer review of "Developing Conversion Factors of LCIA Methods for Comparison of LCA Results in the Construction Sector"

_sustainability, doi:10.3390/su13169016_

Round 1

Reviewer 1 Report

The article proposes conversion factors for LCA impact harmonization and demonstrates its application in the construction sector. According to authors, these conversion factors “of an impact category between two LCIA methods are developed based on regression analysis using the least squares method”.

The author claims it is a “novel” method – however, I do not see any novelty in regression analysis.

Further, the article is based on “wrong” concept and this is the MOST IMPORTANT FEEDBACK to the article from my side.

This so-claimed “novel” concept is not only unsubstantiated but rather unnecessary. It challenges the whole school of “LCA science” that is flourishing in the world in the last 30 years and on which analysis world’s leading multinational companies including Forbes 2000, public bodies, government organizations now depend on. So much so that the international standard organization proposed two INTERNATIOALLY RECOGNIZED standards on LCA about 20 yrs ago based on consensus view of leading LCA scientists of the world.

In addition, the authors showed their little to nil skills on LCA by writing: “The allocation methods of the selected construction materials do not have significant effect on the LCIA results, so we pick the consequent allocation method (“Conseq”) from the three available methods (cut-off, attributional, and consequent) in the Ecoinvent database.”

In my view, there are significant differences between the LCA results among these three approaches. Logically, if there is no difference between the three approaches, then why the LCA scientists have developed these three approaches in the first place by the way?

In my own findings, results vary enormously between the three approaches – by not a few times; but by a few order even in some cases..

Further, the authors wrote consequent allocation method – there is no ‘consequent’ method; rather ‘consequential’ method.. this is a gross/fundamental knowledge based mistake for an article where there is no grammatical mistake

Further, the authors mentioned “it is still not known whether all the LCIA methods can be finally harmonized”. This gives the impression that all the different methods must have to provide the same results to be harmonized and also harmonization between the results is a must. But this view is completely wrong.. All methods are based on established science and some are scientifically more robust; some are less robust.. nothing else.. therefore, harmonization is not possible until the impact model developers agree on a consensus best model.. UNEP/SETAC’s target on harmonization is also not the same as the authors’ claim!

Reviewer 2 Report

There are some weaknesses through the manuscript which need improvement. Therefore, the submitted manuscript cannot be accepted for publication in this form, but it has a chance of acceptance after a major revision. My comments and suggestions are as follows:

1- Abstract gives information on the main feature of the performed study, but some details about the proposed method  must be added.

2- Authors must clarify necessity of the performed research. Objectives of the study must be clearly mentioned in introduction.

3- The literature study must be enriched. In this respect, authors must read and refer to the following papers: (a) LCA in 3D printing: https://doi.org/10.1016/j.apmt.2020.100689 (b) LCA in sustainable development: https://doi.org/10.1016/j.jclepro.2020.122056

4- It would be nice, if authors could add some figures (real or schematic) to show concept and some conditions.

5- The main reference of each formula must be cited. Moreover, each parameters in equations must be introduced. Please double check this issue. Italic wiring of formula must be avoided.

6- In 3.2 was mentioned “two LCIA methods are 147 developed based on regression analysis using the least squares method”. Details of this least square error must be presented.

7-All error in calculation must be considered and discussed.

8- In its language layer, the manuscript should be considered for English language editing. There are sentences which have to be rewritten.

9- The conclusion must be add at the end and it must be more than just a summary of the manuscript (the future opportunities is not enough as last part of the paper). . List of references must be updated based on the proposed papers. Please provide all changes by red color in the revised version.

Reviewer 3 Report

Dear Authors,

Thank you authors for the interesting manuscript with title „Harmonizing the Indicators of LCIA Methods - Development of Conversion Factors for the Construction Sector“ (ID sustainability-1289649).  The research design, questions, hypotheses, and methods are clearly presented. Several comments for the improvement of the manuscript are as follow:

  1. The literature review with analysis of the selected problem: indicators of LCIA can be represented in the manuscript as a review part of the manuscript;
  2. The sustainable aspects can be indicated and present as a part in the text/case study;
  3. A more detailed appraisal with a selected case study can be presented in the part of 3.2 „Case study“;
  4. The comparison results with a required value of each Item can be presented in Table 3.
  5. The literature analysis with a reference list with a new literature source (from the 2020-2021 year) can be presented.

Reviewer

Round 2

Reviewer 1 Report

As the authors said in the abstract: this study “proposes an effective method of conversion factors (CFs) for converting the results of 8 LCIA methods for 14 impact categories and then demonstrates its application in the construction sector.” However, the concept of developing conversion factors between LCIA methods is wrong and not needed. Further, such conversion factor is not universal like 1 mile = 1.60934 km. There are millions of variables that will vary these conversion factors. So that means based on different cases there will be millions of conversion factors. Based on the typical mile to km example, the value 1.60934 will vary from nearly zero to any higher value. That means no conversion factor is correct conversion factor which is not the case for mile to km conversion.

I, therefore, can’t support accepting this article with such gross mistakes. Further, I am astonished by the attitude of the authors who submitted this article 2nd time even after I pointed out these core mistakes once in my 1st review feedback.

Author Response

Pls see the attachment.

Reviewer 2 Report

Dear Authors,

you have addressed the comments, and answered the questions. The revised version of your manuscript appears to be suitable for publication.

Author Response

Thank you very much for your comments!